# Graviton mass and entanglement islands
# in low spacetime dimensions

**Hao Geng**

Center for the Fundamental Laws of Nature, Harvard University,
17 Oxford St., Cambridge, MA, 02138, USA

## Abstract

It has been conjectured and proven that entanglement island is not consistent with long-range (massless) gravity in a large class of spacetimes, including typical asymptotically anti-de Sitter spacetimes, in high spacetime dimensions. The conjecture and its proof are motivated by the observation that existing constructions of entanglement islands in high dimensions are all in gravitational theories where the graviton is massive for which the standard gravitational Gauss' law doesn't apply. In this letter, we show that this observation persists to lower dimensional cases. We achieve this goal by providing a unified description of the gravitational Gauss' law violation in island models that can work in any dimensions. This unified description teaches us new lessons on entanglement islands and subregion physics in quantum gravity. We focus on the case of the (1+1)-dimensional Jackiw-Teitelboim (JT) gravity for the purpose of demonstration.



## 1  Introduction

The AdS/CFT correspondence [1–3] states that quantum gravity in a $(d + 1)$-dimensional asymptotically anti-de Sitter space (AdS$_{d+1}$) duals to a d-dimensional conformal field theory (CFT$_d$), which is often thought to be living on the asymptotic boundary of the AdS$_{d+1}$.

This correspondence has remarkable implications to the black hole information paradox [4,5] as it states that the dynamics and states of a black hole in $AdS_{d+1}$ is fully encoded in a $CFT_d$ which is manifestly unitary. However, to have a proper understanding of the black hole information paradox, we have to have a detailed study of its radiation and this is not easy in the standard AdS/CFT. The difficulty is due to subtleties from the gravitational nature of $AdS_{d+1}$ bulk [6–10]. This observation motivates us to engineer a model in the AdS/CFT correspondence where the radiation from a black hole is under control.

This can be achieved by coupling the black hole spacetime to an external thermal bath. The bath is nongravitational and is described a $CFT_{d+1}$ (see Fig. 1). It extracts the black hole radiation so we can study the radiation by studying the bath. More precisely, we can dualize the $AdS_{d+1}$ black hole to a $CFT_d$ and think of such a model from the pure field theory perspective as putting a $CFT_d$ on the boundary of a $CFT_{d+1}$ and this coupling is realized by a marginal deformation [11,12]

$$S_{\text{tot}} = S_{\text{CFT}_d} + \int \mathrm{d}^d x : \mathcal{O}_1(x)\mathcal{O}_2(x): + S_{\text{CFT}_{d+1}}, \tag{1}$$

where $\mathcal{O}_1(x)$ is a single-trace primary operator in $CFT_d$ and $\mathcal{O}_2(x)$ is the boundary extrapolation of a primary operator of the bath $CFT_{d+1}$. Recent studies show that a low dimensional version of this model [13–16]– the Jackiw-Teitelboim (JT) gravity or higher dimensional versions with holographic dual– the Karch Randall braneworld [10,17–21] enable us to compute the so-called Page curve [22] of the black hole radiation with results consistent with unitarity. In this calculation, a new element in quantum gravity– entanglement island $\mathcal{I}$ emerges, which is a gravitational subregion living in the entanglement wedge of a nongravitational system $\mathcal{R}$ and is however disconnected from $\mathcal{R}$. In the context of an evaporating black hole, the system $\mathcal{R}$ is its early-time radiation and the subregion $\mathcal{I}$ overlaps the black hole interior after the Page time. This suggests that after the Page time the physics in the interior of the black hole is fully encoded in its early-time radiation..

Nevertheless, it has been known that in such models at higher dimensions ($d \geq 3$) the graviton in the $AdS_{d+1}$ is massive and this mass is induced from the bath coupling [11,12,23–26]. This effect has been found to have profound implications to quantum gravity by noticing that the gravitational Gauss' law is modified by the graviton mass [20,27,28]. There exist several ways to understand this mass generating effect from different descriptions and aspects of the set-up. On the one hand, using the AdS/CFT correspondence, the graviton mass is dual to the anomalous dimension of the $CFT_d$ stress-energy tensorand this anomalous dimension is a result of the marginal deformation in Eq. (1) [11,12]. On the other hand, a direct calculation of the one-loop graviton propagator [24,25] shows that the graviton indeed gets a mass due to the bath coupling. Furthermore, the calculation in [24,25] suggests that diffeomorphism symmetry is not manifestly broken in the full quantum gravity theory in $AdS_{d+1}$ but spontaneously broken. As a result, the graviton mass is generated by the Higgs mechanism. The Higgs mechanism for graviton mass requires 1) spontaneously broken diffeomorphism symmetry, 2) a Goldstone vector boson coupled to the graviton in the Stückelberg form and as a result 3) the graviton obtains a mass by eating this vector boson. However, instead of showing 1) and 2) with 3) as a result, [24] showed that there is a composite vector mode induced from the modes of an $AdS_{d+1}$ scalar field due to the bath coupling and it propagates in the graviton self-energy loop generating the graviton mass. The gist of the calculation in [24,25] is that the bulk graviton has a propagator at tree-level and the one-particle irreducible self-energy from the composite vector mode of the bulk scalar field in the loop induces a nonzero pole for the graviton propagator. This makes it difficult to generalize the observation that the graviton is massive which modifies the gravitational Gauss' law to lower dimensions ($d \leq 2$) due to the fact that here the bulk graviton lacks a tree-level propagator. Nevertheless, it is expected that

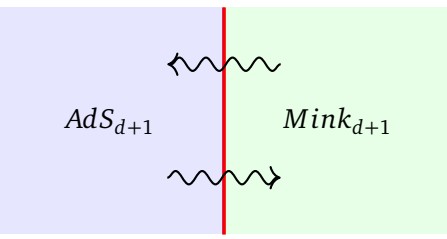

Figure 1: We couple the gravitational $AdS_{d+1}$ with a nongravitational bath by gluing them along the asymptotic boundary the $AdS_{d+1}$. The bath is modeled by a $CFT_{d+1}$ living on a half Minkowski space sharing the same boundary as the asymptotic boundary of the $AdS_{d+1}$.

the modification of the gravitational Gauss' law in the set-up with a bath coupled should persists. Otherwise there would be a sharp contradiction between the existence of entanglement island [16] and the basic requirement of the entanglement wedge reconstruction [28].[1]

In this paper, we focus on the case of (1+1)-dimensional (2d) Jackiw-Teitelboim (JT) gravity studied in [16]. It is shown explicitly that we have entanglement island if we couple the bulk $AdS_2$ spacetime with a nongravitational bath. We will show that in this set-up even though there is no kinetic term for the graviton in $AdS_2$ there will be a graviton mass term induced from the bath coupling which modifies the gravitational Gauss' law.

## 2 Gravitational Gauss' law and entanglement islands in JT gravity

In this section, we explain the gravitational Gauss' law and briefly review the construction of entanglement island in the context of JT gravity. In the end, we discuss why they are in tension in a nutshell.

JT gravity [32,33] is a 2d dilaton gravity theory with the action

$$S = \frac{1}{16\pi G_N} \int d^2x \sqrt{-g}\, \phi(x)(R[g]+2) + S_{\text{matter}}[\psi; g]\,, \tag{2}$$

where $G_N$ is the Newton's constant, $R[g]$ is the Ricci scalar, $\phi(x)$ is the dilaton field and $S_{\text{matter}}[\psi; g]$ is the action of the minimally coupled matter fields in the 2d manifold with metric $g_{\mu\nu}(x)$.

The dilaton is a Lagrange multiplier whose path integral fixes the Ricci scalar to be $R[g] = -2$. Due to the 2d nature of the geometry and diffeomorphism invariance, the metric $g_{\mu\nu}(x)$ is conformally flat so we can consider the $AdS_2$ Poincaré patch

$$ds^2 = g_{\mu\nu}(x)dx^\mu dx^\nu = \frac{1}{x^2}\eta_{\mu\nu}dx^\mu dx^\nu = \frac{-dt^2+dx^2}{x^2}\,, \tag{3}$$

where the asymptotic boundary is at $x \to 0$. All other asymptotic AdS solutions to $R[g] = -2$ are diffeomorphic equivalent to Equ. (3). Hence we can take the metric to be fixed as Equ. (3) even in quantum theory. In JT gravity we usually cut off the geometry at $x = \epsilon$ for $\epsilon \to 0$ and a boundary term in the action is important to ensure the action variation is well-defined [34]. The variation of Equ. (2) with respect to the metric gives an equation of motion

$$8\pi G_N T_{\mu\nu} = -\nabla_\mu\nabla_\nu\phi + g_{\mu\nu}\nabla^2\phi - g_{\mu\nu}\phi\,, \tag{4}$$

---

[1]See [29–31] for string theory evidence that the existence of entanglement island relies on graviton mass.

where $T_{\mu\nu}$ is the matter fields' energy-momentum tensor. Since the geometry is fixed to be Equ. (3), if we linearize the above equation around a vacuum solution $\phi^{(0)}$ (i.e. it satisfies Equ. (4) with $T_{\mu\nu} = 0$) we have

$$8\pi G \delta T_{\mu\nu} = -\nabla_\mu \nabla_\nu \delta\phi + g_{\mu\nu}\nabla^2\delta\phi - g_{\mu\nu}\delta\phi. \tag{5}$$

In the case that there is an energy excitation in the bulk we have

$$\delta\rho = -\delta T_0^0 = -\frac{1}{8\pi G_N}g^{xx}\nabla_x\nabla_x\delta\phi + \frac{1}{8\pi G_N}\delta\phi, \tag{6}$$

which can be written as

$$\delta H_{\text{matter}} = \frac{1}{8\pi G_N}\int dx\sqrt{-g}\left[-g^{xx}\nabla_x\nabla_x\delta\phi + \delta\phi\right], \tag{7}$$

where $H_{\text{matter}}$ denotes the matter Hamiltonian. Using the metric Equ. (3), we can see that

$$\delta H_{\text{matter}} = -\frac{1}{8\pi G_N}\int dx \partial_x\left[\partial_x\delta\phi + \frac{\delta\phi}{x}\right], \tag{8}$$

which shows that local energy excitations in the bulk can be detected by a near boundary measurement of the change in the dilaton field at the same instant of the excitation. This can be uplifted to a quantum mechanical statement that the bulk matter field Hamiltonian operator equals to an AdS$_2$ boundary operator [9, 28, 35, 36]. This is the Gauss's law in JT gravity.

The importance of JT gravity shows up in its recent application to the calculation of the Page curve for black hole radiation [13, 14]. In this calculation, the black hole under consideration is a four dimensional near-extremal charged black hole, for which the s-wave sector of the matter-gravity coupled theory is described by Equ. (2) [37, 38]. To study the radiation from such a black hole, a two-dimensional nongravitational bath is glued to the asymptotic boundary of the reduced geometry Equ. (3). Such a coupling is achieved by modifying the boundary condition of the AdS$_2$ matter fields $\psi(x)$ from reflective to transmissive. This is realized as a marginal deformation in the dual field theory description as in Equ. (1). To simplify the calculation, both the AdS$_2$ matter fields $\psi(x)$ and the nongravitational bath were modeled by a 2d conformal field theory– the free massless fermion in [16], for which the formula of the entanglement entropy is known in a closed form. Now the entropy of the radiation can be studied by calculating the entanglement entropy of a subregion in the bath (see Fig.2). The result is that this entanglement entropy obeys the unitary Page curve and at late times the entanglement wedge [39] of the radiation contains a disconnected region in the AdS$_2$ bulk which is called an *entanglement island*. The holographic interpretation of this result is that the physics in the island is fully encoded in the radiation and is totally independent from its complement $\bar{\mathcal{I}}$ in the bulk AdS$_2$.

From here we can see that there is a direct contradiction between the physics of the entanglement island and the gravitational Gauss' law Equ. (8). The reason is that a near-boundary observer in AdS$_2$ can detect the existence of an energy excitation inside the island by measuring $\frac{1}{8\pi G_N}\left(\partial_x\delta\phi(x) + \frac{\delta\phi(x)}{x}\right)$ at the same instant of the excitation.[2] Hence the physics in the island is not independent of its complementary region in the bulk AdS$_2$.

---

[2]This statement can be uplifted to a quantum mechanical statement that there exists an AdS$_2$ boundary operator which doesn't commute with a local unitary operator in the island region which is made from the matter fields. This is due to the gravitational Gauss' law Equ. (8) which states that the matter Hamiltonian equals to an AdS$_2$ boundary operator.

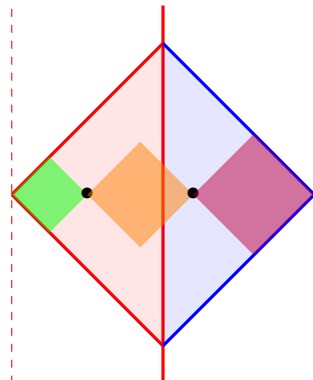

Figure 2: The demonstration of the island calculation in [16]. We draw a Penrose diagram of the set-up where the red shaded region is the bulk AdS$_2$ described by geometry Equ. (3), the blue shaded region is the bath whose geometry is half of a 2d Minkowski space and the bath is glued to the bulk along the thick red vertical line ($x = 0$). The pink causal diamond in the bath models the radiation $\mathcal{R}$ and the green causal diamond in the AdS$_2$ is the island $\mathcal{I}$ of $\mathcal{R}$. The physics in the island should be independent of that in the orange diamond.

## 3 Gravitational Gauss' law violation in the JT island model

In this section, we model the bulk matter field by a massive scalar field $\psi$ with mass square $m^2$ for which the transparent boundary condition with the bath glued on can be explicitly written down as the holographic dual of the marginal deformation Equ. (1) [12,24,26]. Let's consider the full gravitational path integral in the gravitational sector of the system

$$Z_{\text{Grav}} = \int D[\phi]D[g_{\mu\nu}]D[\psi]e^{-iS_{\text{JT}}[\phi;g]-iS_{\text{matter}}[\psi;g]}, \qquad (9)$$

where we can treat the metric fluctuation perturbatively and split it into two parts– a background part and a perturbative fluctuation

$$g_{\mu\nu}(x) = g^0_{\mu\nu}(x) + \sqrt{16\pi G_N}h_{\mu\nu}(x), \qquad (10)$$

where $h_{\mu\nu}(x)$ is symmetric traceless. In our perturbative calculation, we have to choose a consistent background for the metric and dilaton $(g^0_{\mu\nu}(x), \phi^0(x))$. We take the background to satisfy the vacuum (zero matter source) equation of motion. That is the metric is Equ. (3) and the dilaton profile $\phi^{(0)}$ satisfies Equ. (4) with $T_{\mu\nu} = 0$. Hence we only have to consider the matter part and its coupling with $h_{\mu\nu}$ and the dilaton fluctuation $\delta\phi(x)$. Let's focus on the $h_{\mu\nu}$ part for now. To the leading order in $G_N$ we have

$$Z_h = \int D[\psi]D[h_{\mu\nu}]e^{-iS_{\text{matter}}[\psi;g^0]-iS_{\text{int}}[\psi,h;g^0]},$$

$$S_{\text{int}}[\psi,h;g^0] = \sqrt{16\pi G_N}\int d^2x\sqrt{-g^0(x)}h_{\mu\nu}(x)T^{\mu\nu}_{\text{matter}}(x). \qquad (11)$$

As we will show later, in this system the diffeomorphism symmetry

$$h_{\mu\nu}(x) \to h_{\mu\nu}(x) + \nabla_\mu\epsilon_\nu(x) + \nabla_\nu\epsilon_\mu(x), \qquad (12)$$

is spontaneously broken due the the fact that the stress-energy two-point correlator is not divergence free.

Nevertheless, it will be useful for us to consider this system as a gauge-fixed version of a fully diffeomorphism invariant system. This can be realized by introducing a divergenceless Stückelberg vector field $V^\mu(x)$ which transforms under the diffeomorphism Equ. (12) as

$$V^\mu(x) \rightarrow V^\mu(x) - \epsilon^\mu(x), \tag{13}$$

and we define

$$
\begin{aligned}
h'_{\mu\nu}(x) &= h_{\mu\nu}(x) + \nabla_\mu V_\nu(x) + \nabla_\nu V_\mu(x) \\
&\equiv h_{\mu\nu}(x) + \nabla_{(\mu} V_{\nu)}(x).
\end{aligned}
\tag{14}
$$

We can construct the partition function of the fully diffeomorphism invariant system by integrating in the field $V^\mu(x)$

$$
\begin{aligned}
Z_{\text{full}} &= \int D[V_\mu] D[\psi] D[h_{\mu\nu}] e^{-iS_{\text{matter}}[\psi;g^0] - iS_{\text{int}}[\psi,h;g^0]} \\
&= \int D[V_\mu] D[\psi] D[h'_{\mu\nu}] e^{-iS_{\text{matter}}[\psi;g^0] - iS_{\text{int}}[\psi,h';g^0]} \\
&= \int D[V_\mu] D[\psi] D[h_{\mu\nu}] e^{-iS_{\text{matter}}[\psi;g^0] - iS_{\text{int}}[\psi,h';g^0]},
\end{aligned}
\tag{15}
$$

which is manifestly invariant under the transformation Equ. (12) and Equ. (13) and fixing the gauge by setting $V^\mu(x) = 0$ we get back to Equ. (11).

To understand the physics in the graviton sector, we can integrate out the matter field $\psi(x)$ in Equ. (15) getting an effective action $S_{eff}[h,V]$ which has to be a function of the diffeomorphism invariant combination Equ. (14). Therefore, we can focus on the $V^\mu(x)$-sector and at the end replace $\nabla_\mu V_\nu(x) + \nabla_\nu V_\mu(x)$ by the symmetric combination Equ. (14). Since we want to understand the low energy physics, we only care about the leading order effective action for $V^\mu$ in $G_N$ expansion which is

$$S_{\text{eff}}[V] = -i \frac{16\pi G_N}{2} \int d^{d+1}x \sqrt{-g_0(x)} d^{d+1}y \sqrt{-g_0(y)} \nabla_{(\mu} V_{\nu)}(x) \Pi^{\mu\nu,\rho\sigma}(x,y) \nabla_{(\rho} V_{\sigma)}(y). \tag{16}$$

In our case $d = 1$ and we have

$$\Pi^{\mu\nu,\rho\sigma}(x,y) = \left\langle T\left( T^{\mu\nu}_{\text{matter}}(x) T^{\rho\sigma}_{\text{matter}}(y) \right) \right\rangle. \tag{17}$$

The low energy physics is encoded in the long distance limit that $x$ and $y$ are widely separated. It has been calculated in detail in [12, 24, 25] that in large distance regime the result is given by

$$16\pi G_N \Pi^{\mu\nu,\rho\sigma}(x,y) = 2M^2 \nabla^\mu D^{\nu,\sigma}(x,y) \nabla^\rho, \tag{18}$$

where $D^{\nu,\sigma}(x,y)$ is in the form of the propagator of a divergenceless vector field satisfying

$$(\nabla^2 - d) D^{\nu,\sigma}(x,y) = -i \frac{g^{\nu\sigma}}{\sqrt{-g}} \delta^{d+1}(x-y), \tag{19}$$

and $M^2$ is given in general in [12] as

$$M^2 = -G_N \frac{2^{4-d} \pi^{\frac{3-d}{2}}}{(d+2)\Gamma(\frac{d+3}{2})} \frac{a_{\Delta_1} a_{\Delta_2} \Delta_1 \Delta_2 \Gamma[\Delta_1] \Gamma[\Delta_2]}{\Gamma[\Delta_1 - \frac{d}{2}] \Gamma[\Delta_2 - \frac{d}{2}]}, \tag{20}$$

where $a_{\Delta_1}$ and $a_{\Delta_2}$ are of opposite signs[3] and we have

$$\Delta_1 = \frac{d}{2} + \sqrt{\frac{d^2}{4} + m^2}, \quad \Delta_2 = \frac{d}{2} - \sqrt{\frac{d^2}{4} + m^2}. \tag{21}$$

Integrating by parts in Equ. (16) we can see that in the low energy regime we would have

$$\begin{aligned} S_{\text{eff}}[V] &= iM^2 \int d^{d+1}x\sqrt{-g_0(x)}d^{d+1}y\sqrt{-g_0(y)}V^\nu(x)(\nabla^2 - d)D_{\nu,\sigma}(x,y)\nabla_\rho\nabla^{(\rho}V^{\sigma)}(y) \\ &= -M^2 \int d^{d+1}x\sqrt{-g_0(x)}V_\mu(x)\nabla_\rho\nabla^{(\rho}V^{\mu)}(x) \\ &= \frac{M^2}{2} \int d^{d+1}x\sqrt{-g_0(x)}\nabla_{(\mu}V_{\nu)}(x)\nabla^{(\nu}V^{\mu)}(x), \end{aligned} \tag{22}$$

where we integrated by parts in the first and last steps and we used Equ. (19) in the second step. These integration by parts are justified as the asymptotic fall off of the vector field $V_\mu(x)$ is

$$V_\mu(x) \sim z^{d+1}, \tag{23}$$

due to the fact that there is no boundary source of $V_\mu(x)$ [26].

Hence the total low energy effective action is

$$S_{\text{eff}}[h,V] = \frac{M^2}{2} \int d^{d+1}x\sqrt{-g_0(x)}h'_{\mu\nu}(x)h'^{\mu\nu}(x). \tag{24}$$

If we fix the diffeomorphism gauge Equ. (12) and Equ. (13) by setting $V^\mu = 0$ we have

$$S_{\text{eff}}[h] = \frac{M^2}{2} \int d^{d+1}x\sqrt{-g(x)}h_{\mu\nu}h^{\mu\nu}, \tag{25}$$

which is nothing but a mass term for graviton. Here we notice that this calculation works in any dimensions and it gives the graviton mass term without the requirement of having a tree level graviton propagator.

Now we can show that the gravitational Gauss' law is modified. The reason is that with the "graviton mass" term incorporated the full Einstein's equation Equ. (4) is modified to

$$8\pi G_N T_{\mu\nu} = -\nabla^{(0)}_\mu\nabla^{(0)}_\nu\phi + g_{\mu\nu}\nabla^{(0)2}\phi - g^{(0)}_{\mu\nu}\phi - \frac{8\pi G_N M^2}{2}h_{\mu\nu}, \tag{26}$$

where we use $\nabla^{(0)}$ to emphasize that the covariant derivative is taken in the background geometry Equ. (3). Now as we did in Equ. (5) we can linearize this equation around the background profile $\phi^{(0)}$ which gives us

$$8\pi G_N \delta T_{\mu\nu} = -\nabla^{(0)}_\mu\nabla^{(0)}_\nu\delta\phi + g_{\mu\nu}\nabla^{(0)2}\delta\phi - g^{(0)}_{\mu\nu}\delta\phi - \frac{8\pi G_N M^2}{2}h_{\mu\nu}, \tag{27}$$

from which we get

$$\delta H_{\text{matter}} = -\frac{1}{8\pi G_N} \int dx\partial_x\Big[\partial_x\delta\phi + \frac{\delta\phi}{x}\Big] + \frac{M^2}{2} \int d^2x\sqrt{-g}h_0^0, \tag{28}$$

which is not just a boundary term. As a result, a bulk energy excitation can be hidden from being detected by the observer near the asymptotic boundary due to the graviton mass. This is exactly the same result as in the higher dimensional cases discovered in [28, 40]. This is now consistent with the existence of entanglement island as in Fig.2.

---

[3]They are both nonzero in bath coupled case and either one of them is zero when there is no bath [26].

# 4 Conclusions

In this paper, we finished the program initiated in [20] to prove that gravitational theories in the island models are massive gravity theories by proving this conjecture in low dimensional cases. We mainly focused on the JT gravity model where the island model can be analytically solved [16]. To do this we developed a tool to efficiently extract the graviton mass Equ. (20). Nevertheless, we emphasize that the tool we developed provides a generic proof of the conjecture in any dimensions and concretely realizes the fact that the diffeomorphism symmetry is spontaneously broken by the bath coupling. Our study can be generalized to the general higher spin gravity for which the bath breaks the conservation of higher spin currents and we can efficiently show that the higher spin symmetries are spontaneously broken and the corresponding higher-spin gauge bosons are massive [41]. Further discussions of the implication of our work are provided in the Supplemental Material.

# Acknowledgments

We are grateful to Amr Ahmadain, Luis Apolo, Jan de Boer, A. Liam Fitzpatrick, Åsmund Folkstad, Eduardo Gonzalo, Brianna Grado-White, Daniel Harlow, Andreas Karch, Indranil Halder, Ami Katz, Matthew Heydeman, Yangrui Hu, Yikun Jiang, Juan Maldacena, Geoff Penington, Carlos Perez-Pardavila, Lisa Randall, Suvrat Raju, Martin Sasieta, Pushkal Shrivastava, Brian Swingle, Neeraj Tata and Gabriel Wong for useful discussions at various stages of this project. We thank Andreas Karch, Carlos Perez-Pardavila, Lisa Randall, Suvrat Raju and Marcos Riojas for comments on an early version of the draft.

**Funding information** The work of HG is supported by the grant (272268) from the Moore Foundation "Fundamental Physics from Astronomy and Cosmology" and a grant from Physics Department at Harvard University.

# A Supplemental material

In this supplementary section, we discuss further implications of our work.

Our study in fact conveys important lessons on entanglement islands and subregion physics in gravity. It is argued in some papers [42, 43] that the tension between the entanglement island and the long-range nature of gravity discovered in [20, 27, 28] can be avoided if we consider background configurations with strong enough features. The essential observations in these papers are that 1) we can dress local bulk operators to these features to avoid dressing to the asymptotic boundary or equivalently 2) in these background configurations the gravitational Gauss' law would be significantly modified, for example due to the lack of a globally defined time-like Killing vector. From 2) we can see that these are essentially different ways to break the long-range nature of gravity. Nevertheless, these ways are highly non-universal as they would propose that entanglement islands can exist only in very specific backgrounds and most of the existing models of entanglement islands don't satisfy this corollary. Such non-universality also manifests by the consideration that at late times these background with nontrivial matter distribution will collapse to form a black hole which looses such features due to the no hair theorem. Meanwhile, papers like [42, 43] are in tension with entanglement wedge reconstruction [39] which states that operators inside island $\mathcal{I}$ should nonperturbatively commute with those from its complement $\bar{\mathcal{I}}$. On the other hand, all existing models of entanglement island, which is a closed subregion inside a gravitational universe, are in the

context of massive gravity due to the bath coupling [20,26]. This observation highly motivates us to believe that there is a universal connection between entanglement islands and massive gravity. The essential reason is that massive gravity provides a global feature to dress operators diffeomorphism invariantly and hence entanglement islands can exist in any backgrounds in this context. This can be seen in the fully covariant description of massive gravity we outlined in this paper and [26], where the dynamics of the Stückelberg vector field manifests, by noticing that the Stückelberg field also transforms under diffeomorphism and so we can use this field to dress operators diffeomorphism invariantly. For example, to the first order in $G_N$ expansion which is the order we are focusing on in this paper, a scalar operator $\hat{\phi}(x)$ can be dressed diffeomorphism invariantly as

$$\hat{\Phi}(x) = \hat{\phi}\left(x + \sqrt{16\pi G_N}V\right), \tag{A.1}$$

and higher order dressings can be easily done order by order as in [6, 7]. Interestingly, in the gauge-fixed description ($V^\mu = 0$) this dressing disappears, i.e. all diffeomorphism invariant operators becomes local, and meanwhile the gravitational Gauss' law violation manifests. From here we can see that information localizes differently in massive gravity theories compared to the massless gravity theories [40], as in massive gravity information can be perturbatively hidden in the bulk. This is the essential reason why the existence of entanglement islands is universally allowed in massive gravity theories.

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
