# Peer review of "Graviton Mass and Entanglement Islands in Low Spacetime Dimensions"

_SciPost Physics, doi:SciPost Phys. 19, 146 (2025)_

## Round 2 · Referee Report · Anonymous (Referee 1) · 2025-8-8

Report
Recommendation
Publish (surpasses expectations and criteria for this Journal; among top 10%)

---

## Round 2 · Referee Report · Anonymous (Referee 2) · 2025-10-11

This paper makes a timely contribution to our understanding of entanglement islands in quantum gravity by extending the islands-massive gravity connection from higher dimensions to (1+1)-dimensional Jackiw-Teitelboim (JT) gravity. The author addresses a genuine puzzle that has remained unresolved since the original island calculations in JT gravity: how can entanglement islands be consistent with gravitational Gauss' law, which would seemingly allow boundary observers to detect energy excitations deep inside the island region?

The central achievement is the understanding of gravitational Gauss' law violation that works across all spacetime dimensions, completing the program initiated by previous papers. The technical innovation lies in efficiently extracting the graviton mass without requiring a tree-level graviton propagator, which is crucial since JT gravity lacks a kinetic term for the graviton. The author accomplishes this through a Stückelberg formulation. This approach also clarifies that diffeomorphism symmetry is spontaneously broken by the bath coupling, with the mass arising through a Higgs-like mechanism in the graviton self-energy loop.

The result has significant implications in illustrating the entanglement islands. By demonstrating how the graviton mass modifies Einstein's equations in JT, the author resolves the tension with island factorization and establishes a unified approach to show that entanglement islands exist only in massive gravity theories. The explicit mass formula in JT connects naturally to earlier higher-dimensional results, providing crucial theoretical foundation for more recent papers in double holography and EOW branes where bath coupling universally generates massive gravitons.

One question that deserves further thinking is about the physical interpretation of graviton mass in JT gravity. Since there are no propagating gravitons in two dimensions and the dilaton is the only dynamical degree of freedom, it would be illuminating for the author to further elaborate on how the mass term should be understood in relation to the dilaton's phase and how the mass scale relates to $AdS_2$ scales. This conceptual clarification would further strengthen an already excellent work. One may also wonder that since in this example, a consistent coupling to a system that can absorb radiation leads to a modification of long-range graviton dynamics, whether this suggest that in other examples of open (holographic) quantum systems with gravity can also be effectively described by massive gravity.

The manuscript is well written and addresses an important gap in our understanding of islands. The author has successfully proven that the islands-massive gravity conjecture holds universally across all spacetime dimensions, providing essential consistency checks for the island formula and entanglement wedge reconstruction in quantum gravity. This work represents a significant advance in resolving the black hole information paradox and deserves prompt publication. I recommend publication in SciPost Physics.

---

## Editorial Decision

published